# Nutritional Quality of Hidden Food and Beverage Advertising Directed to Children: Extent and Nature of Product Placement in Mexican Television Programs

**DOI:** 10.3390/ijerph17093086

**Published:** 2020-04-29

**Authors:** Ana Munguía-Serrano, Lizbeth Tolentino-Mayo, Florence L. Théodore, Stefanie Vandevijvere

**Affiliations:** 1Center for Nutrition and Health Research, National Institute of Public Health, Cuernavaca, Morelos 62100, Mexico; cinys32@insp.mx (A.M.-S.); ftheodore@insp.mx (F.L.T.); 2El Colegio de Chihuahua, Partido Díaz 4723, Progresista, Ciudad Juárez, Chihuahua 32310, Mexico; 3School of Population Health, The University of Auckland, Auckland 1142, New Zealand; s.vandevijvere@auckland.ac.nz

**Keywords:** product placement, children obesity, television, advertising, food and beverages, extent and nature, nutritional content, nutrient profile models, public health

## Abstract

(1) Background: Nutrient-poor, energy-dense food and beverage (F&B) advertisements influence children’s food preferences, consumption, and purchase requests, contributing to overweight and obesity. Objective: To characterize the nutritional quality of F&B advertised by product placement (PP) in Mexican television programs with the highest audience ratings for children. (2) Methods: A total of 48 h of television programs between December 2016 and January 2017 during the hours with the highest ratings for children were analyzed. Nutritional quality was assessed through the Mexican Ministry of Health (MMH-NPM), the World Health Organization Regional Office for Europe (WHO-Europe), and the Pan American Health Organization nutrient profile models (PAHO-NPM). (3) Results: A total of 119 F&B were broadcast, of which more than 60% were unhealthy according to the three nutritional models. Reality shows and movies presented the most PP advertising. The food category most frequently advertised was sugar-sweetened beverages (41.2%). F&B advertised in children’s programs had a higher content of energy, total fat, and saturated fat (*p* < 0.01). (4) Conclusion: The MMH-NPM was the most permissive and the PAHO-NPM was the strictest for evaluating nutritional quality. Mexico must strengthen the regulation of advertising to protect children from its negative effects on health.

## 1. Introduction

Currently, overweight and obesity in children represent an important public health issue around the world [1]. Nearly one in six children are overweight or obese in the Organization for Economic Cooperation and Development (OECD) countries and these rates have increased since 2000 [2].

Mexico is a country with an alarming prevalence of overweight and obesity, with rates of 35.6% for children aged 5–11 years and 38.4% for adolescents aged 12–19 years [3].

According to recent studies, children’s exposure to obesogenic environments is one of the most important determinants of unhealthy diets and energy overconsumption [4,5]. These obesogenic environments are characterized by: the absence of access to recreational or sports facilities; the presence of motorized transport over active transportation options; the absence of green spaces or parks and sidewalks [6]; and the widespread availability of cheap, highly palatable, nutrient-poor, energy-dense food, including sugar-sweetened beverages (SSB), that is also heavily promoted through many media channels, such as television [7].

Additionally, the prevalence of obesity in children is positively correlated with the number of hours spent viewing television due to greater periods of inactivity and exposure to nutrient-poor, energy-dense food and beverage (F&B) advertising [8,9]. In 2014, the Federal Institute of Telecommunications in Mexico (IFT, for its acronym in Spanish) reported that Mexico is the country with the highest prevalence of television viewing hours by children (4:34 h per day), followed by USA (3–4 h), Peru (3:30 h), Colombia (2:45 h), Italy (2:42 h), Spain (2:38 h), and France (2:18 h) [10]. In addition, 77.3% of children and 78.6% of adolescents in Mexico spent more than 2 hours a day in front of a screen [11], which included time spent viewing television, movies, and videos, browsing the internet and social and online media, or playing video games, whereas the recommendation of the American Academy of Pediatrics is to limit children’s total media time (with entertainment media) to a maximum of 1 to 2 h of quality programming per day [12].

Evidence has previously found that the F&B most frequently promoted to children are high in saturated fat, sugar, or sodium [13]. Furthermore, numerous studies have shown that nutrient-poor, energy-dense food advertisements influence children’s food preferences, consumption patterns, and purchase requests [13,14]. In addition, the food industry uses persuasive marketing techniques, such as attractive product packaging, toys, and emotional appeals, to create links with children and loyalty with the brand [5].

Studies conducted in Mexico have shown that more than 60% of F&B advertised on national television did not meet any nutritional quality standards [15] and about 75% of F&B advertisements were targeted at children [16]. Despite the existing evidence of the extent and nature of nutrient-poor, energy-dense F&B advertising in Mexico, there are still evidence gaps, including product placement (PP).

According to the World Health Organization (WHO), product placement is a marketing technique that uses a message, brand logo, or product (food or beverage) in a visual or graphic medium, in a variety of forms of media entertainment, including television programs, films, music, and video games [17]. The main difference between PP and TV advertisements is the insertion of products or brands within the program content (movies, shows, and series) [18]. Brands are intentionally incorporated into content in a visible way to persuade the audience [19], for example, when a principal character of a movie or series is shown drinking a soft drink of a specific brand during a scene, such as Coca-Cola or Sprite.

The entertainment industry is growing and is now distributed and consumed online, by mobile phones, on TV, on streaming platforms (such as YouTube, Netflix, and Amazon Prime Video), or in cinemas. These changes have opened the door to PP [20] and there are ethical concerns over its use because of its deception and the “subliminal” effect [21]. Studies have shown that people recall strategic PP four times as much as commercial messages, suggesting that PP can be far more effective than 30-second TV commercials [20].

There is also scientific evidence on PP’s effects on children. Auty and Lewis [22] explored children’s choices after PP exposure (Pepsi Cola) in a movie and found a significant difference between the two groups; children who had seen the branded clip preferred the soda placed in the movie. The authors also concluded that previous exposure together with a reminder in the form of recent exposure affects children´s choice [22]. Hudson and Elliot [20] found that after F&B PP exposure on TV, children had a strong spontaneous recall for the products placed, especially for the nutrient-poor, energy-dense products (71.3% Pepsi, 58.3% Reese’s, 54.8% Fruit Gushers (fruit candy snacks), and 49.6% Cheetos) and older children (aged 10–12) tended to recall more brands than younger ones (aged 7–9) because of their higher information processing skills [20]. Van Reijmersdal et al. [23] found that interactive PP in video games resulted in more positive attitudes toward the game, higher top-of-mind awareness of the brand, more positive brand images, and more favorable behavioral intentions. In addition, Sutherland [24] found that a large number of movies targeted at children featured PP and the majority of placements were for nutrient-poor, energy-dense F&B.

In Chile, Uribe and Fuentes-Garcia [25] found that brand awareness and the behavioral disposition toward junk food and McDonald’s increased when children were exposed to PP (in comparison with the control group) [25]. Older children (12–15 years old) performed better in brand awareness, but scored lower in behavioral disposition than the 9-year-old group [25]. Furthermore, the synergy of PP with TV advertisements increased the effect and power of these communication techniques [25]. Moreover, evidence suggests that children are not fully aware of the persuasive intent of food marketing and often accept advertising as accurate and truthful, making them a vulnerable target [14,26].

In 2014, the Mexican government regulated the advertising of F&B directed to children younger than 12 years of age that is broadcast on TV and shown in cinemas during specific hours, according to the nutritional criteria proposed by the Ministry of Health in Mexico (MHM). The regulation applies from Monday to Friday between 14:30 h and 19:30 h and weekends from 07:00 h to 19:30 h [27]. The regulated hours are not aligned with Mexican children´s prime time, which is from 19:00 h to 22:00 h, leaving them exposed to the advertising of low nutritional quality products [10,27]. Nutritional criteria do not apply to those programs targeted at audiences of 12 years and older, such as soap operas, sports programming, news, reality shows, and TV series [27], which fails to consider that these programs are among Mexican children’s preferred genres, even if they are not directed to them [10,27]. In addition to these limitations, the current Mexican regulation does not cover PP.

Information regarding PP in Mexican TV programs is unknown; we did not find any scientific publications. Therefore, the aim of this study was to characterize the nutritional quality of F&B advertised by PP in Mexican television programs with the highest child audience ratings through the MMH-NPM, the WHO-Europe, and the PAHO-NPM, according to the target audience, type of PP, program genre, and season. This study will produce evidence to strengthen the present regulation.

## 2. Materials and Methods

### 2.1. Study Design and Sample

In the present cross-sectional exploratory study, a total of 48 h of TV programs were analyzed between December 2016 and January 2017 on the four broadcast open television channels with the highest national audience: XHTV Channel 2, XHGC Channel 5, XHIMT Channel 7, and XHDF Channel 13 [10], during the hours with the highest children audience, that is, 19 h to 22 h (3 h) [10]. We randomly selected one weekday and one day of the weekend for the holiday season and the school season. These days are described in Table 1.

First, for the analysis of the data, the programs to be broadcast in the selected hours and days were identified. Second, the programs were viewed and analyzed in international streaming platforms, such as YouTube and Netflix, as well as in national platforms, such as Blim and the channels’ official websites. One researcher with access to the platforms individually analyzed each program and this was validated by the research leader. Each program was repeatedly watched and analyzed in order to collect all the products. In the case that the same product appeared in several scenes, it was counted each time.

During the 48 h analyzed, 41 programs were aired, but two of them were not found on streaming platforms, ending with a sample of 39 programs to be analyzed. The PP was registered when a branded food or drink appeared during the transmission of the program. PP was divided into: verbal placement when a product/brand was openly discussed; active placement when the product was being used by one of the characters or actors; and passive placement when the product/brand was captured in a shot during the film or television show [18].

### 2.2. TV Program Information

Television programs were categorized by the season in which they were transmitted, considering holiday season from December 17th of 2016 to January 1st of 2017 and school season from January 2nd to January 17th of 2017, according to the 2016–2017 calendar of the Mexican Ministry of Public Education [28]. The genre of the program was categorized into soap operas, dramatic series, series, movies, reality shows, contests, entertainment shows, talk shows, sports, news, food TV shows, and cartoons according to the classification of the IFT [10]. The target audience of the program was determined in accordance with the Mexican Federal Law of Radio and TV. The law categorizes the media content according to the presence of violence, sexuality, addictions, and language, and groups the degree of exposure of these elements into six categories (A, AA, B, B15, C, and D). We categorized children audiences with the A and AA classifications (suitable for all audiences) and general audience with the B, B15, C, and D classifications (directed to adolescents and adults) [29]. The Mexican Federal Law of Radio and TV defines “children” as those aged 12 years or younger. Our study focuses on publicity of food and beverages aimed at children up to 12 years because this is how the Mexican Federal Law of Radio and TV defines this population. 

### 2.3. Product Information

PP advertising can be found in two different forms of presentations: as a product (food and beverage) or as a brand logo. Advertised products were grouped into nine food categories based on the nutrient profile model of the WHO-Europe [30]: savory or sweet snacks, sugar-sweetened beverages (SSB), cereals and grains, dairy products, ready-made food; sauces, dips, and dressings; fresh and frozen meat, poultry, fish, and similar; processed fruit, and vegetables and legumes.

In order to classify the brand logos (without a food or beverage) into food categories and evaluate the nutritional quality, the best-selling products of each brand were identified and categorized. 

Nutritional information was obtained from the nutritional declaration of the labels and the official company websites. In case the nutritional information could not be found, the USDA Food Composition Databases were used [31]. Content of energy, total sugar, fiber, sodium, total fat, and saturated fat was analyzed per 100g/ml. In the case that one product had more than one flavor or the brand logo had different products in the same food category, the average nutritional value was calculated.

The nutritional quality was assessed through three different nutrient profile models: the MMH-NPM [27], the WHO-Europe [30], and the PAHO-NPM [32]. Each product was evaluated separately, with each nutrient profile model, and tagged as “healthy” when the product complied with the specific nutrient criteria and “not healthy” when it did not comply with the nutrient criteria.

### 2.4. Statistical Analysis

The statistical analysis was performed using Stata statistical software package version 14.0. A Mann–Whitney U test to compare median values of the nutritional content by season, type of PP, and target audience was performed. To determine statistically significant differences in the median values of the nutritional content with the genre of the program, the Kruskal–Wallis test and Bonferroni correction were used considering a *p* value < 0.05 as statistically significant. In order to compare which profile model better classified the unhealthy products, a two proportions differences test was used.

## 3. Results

### 3.1. Extent and Nature of Programs and Food and Beverages Advertised by Product Placement

In the present study, 39 TV programs were analyzed, and from these, 23.1% were targeted at children and 25.6% had PP advertising as is shown in Table 2.

A total of 119 products (F&B) were advertised during the programs. More products were advertised during the holiday season than in the school season, 61.3% and 38.7%, respectively, accounting for an average of four products per program during vacation and two products per program during the school season (Table 3). The highest percentage of products advertised was displayed in movies (38.7%), followed by reality shows (36.1%) and sports programs (21%); notably, 74.8% of the products advertised were broadcast in programs targeted at children. The highest percentage of products advertised was SSB (41.2%), followed by cereals (15.1%), savory/sweet snacks and dairy products (9.24%). In this research, only two types of PP were found; 75.6% of F&B were advertised by passive placement and 24.4% by active placement.

### 3.2. The Nutritional Content of Foods and Beverages Advertised

The median energy and nutrient content of the F&B advertised is shown in Table 4. Statistically significant differences were observed in energy and nutrient content according to the season, genre, and target audience. F&B advertised during the holiday season contained significantly higher amounts of energy (214.5 kcal, *p* < 0.001), total fat (2.1 g, *p* < 0.001), and saturated fat (0.14 g, *p* < 0.001). Products advertised during the reality shows were higher in energy (259.4 kcal, *p* < 0.001), total fat (2.7 g, *p* < 0.001), and saturated fat (0.33 g, *p* < 0.001) compared to sports programs, and higher in energy (*p* < 0.001) and total fat (*p* < 0.001), compared to movies. Programs targeted at children contained products with higher amounts of energy (214.5 kcal, *p* < 0.001), total fat (2.1 g, *p* < 0.001), and saturated fat (0.14 g, *p* < 0.001) compared to programs that were not targeted at them.

### 3.3. Nutritional Quality of Foods and Beverages Advertised

More than 60% of the F&B advertised by PP on Mexican television did not comply with any of the three nutrient profile standards (Table 5); 62.2% of the products advertised did not comply with the MMH-NPM, 66.4% with WHO-Europe standards, and 77.3% with PAHO-NPM. The most advertised category, SSB, comprised more than 80% of unhealthy products; 81.6% did not comply with the MMH-NPM and 100% with WHO-Europe and PAHO-NPM. More than 50% of snacks, sauces, dips, dressings, and processed fruit, vegetables, and legumes did not comply with any of the three profile standards. Further, 100% of the category meat, poultry, fish, and similar was classified as healthy and 100% of the ready-made food category as not healthy by the three nutrient profile models. In all the categories, PAHO-NPM evaluated the products more strictly. There were statistically significant differences between the MMH-NPM and PAHO-NPM (*p* < 0.05).

## 4. Discussion

This is the first study to examine PP advertisements in Mexican television programs. This research found that more than 60% of F&B advertised by PP were considered unhealthy according to three nutritional profile models. We found that the F&B advertised during holidays contained significantly higher amounts of energy, total fat, and saturated fat. In addition, the F&B advertised in programs directed to children contained significantly higher amounts of energy, total fat, and saturated fat. The main food category that was advertised was SSB, while the genres that had more PP advertisement were movies and reality shows.

On one hand, there are few studies on the characterization of PP directed to children. Studies in the United Kingdom (UK) and Ireland found slightly less alarming results; they found that 47.5% of foods and 25% of SSBs advertised with PP were considered unhealthy [8]. The genre with the most PP were cartoons, animated programming, tween programming, and movies [33] and the most frequent food category advertised was SSB [8].

On the other hand, much evidence exists regarding the characterization of commercials during schedules with the highest children ratings. In countries such as Mexico [15,34], Argentina [35], and New Zealand [36], more than 60% of the F&B advertised during these schedules were unhealthy and higher in critical nutrients (sodium, fat, and sugar), compared to programs not directed to children [15,34,35,36]. Regarding food categories, in Mexico and the United States of America (USA), SSBs were shown to be the most advertised category during children prime time, as we found in the present study [15,37]. These findings are particularly important to the Mexican context because SSBs contribute 69% of intake from added sugars in the population and their consumption is associated with weight gain and diabetes [38], an important public health problem in Mexico [39,40].

As far as type of program is concerned, movies were the principal genre in which PP advertisement of energy-dense, nutrient-poor F&B was used. Also, the analysis of the prevalence of F&B advertised within the 20 most-viewed movies in the USA for each year from 1996 to 2005 showed that 69% contained at least one food or drink brand [24]. Another relevant genre was reality shows, probably because it is the genre with the second highest children audience in Mexico [10]. There is evidence that this genre mainly uses beverage brand placement to advertise to young people [41]. Finally, regarding season, in Mexico, F&B commercials during the school season contained significantly higher amounts of energy, sugar, total fat, saturated fat, and sodium compared to the holiday season [15], contrary to our findings on PP advertising. This can be explained by the changes that occur in television programming over the holiday season and despite not having evidence of the exposure to children during the holiday season, we assume that they could be more exposed to PP in this period because they have more free time to watch TV.

Our results demonstrate how PP is utilized in food and beverage advertising in Mexican TV. This new marketing strategy may have been introduced to strengthen product promotion and advertise them indirectly, working around the restrictions imposed by Mexican regulations. Furthermore, PP and traditional TV commercials have similar effects on brand awareness and it has been shown that using these strategies together further increases the effect and power of marketing [25]. However, more evidence of how PP was used before the Mexican regulation was implemented is needed to confirm this.

Another important finding of the study is that MMH-NPM is lax and permissive. Another study in Mexico [15] found similar results; Rincón-Gallardo Patiño et al. [15] compared the nutritional quality of F&B advertised on Mexican television, through the MMH-NPM, WHO-Europe, and UK Profile models, finding a percentage of 64.3%, 83.1%, and 78.7% of unhealthy F&B, respectively [15]. Other studies conducted evaluations according to the PAHO-NPM, WHO-Europe, and WHO Eastern Mediterranean nutrient profile models and found that PAHO-NPM had the most strict criteria [32]. It is important to highlight that PAHO-NPM criteria were created by a group of experts and recognized authorities in the field of nutrition with the support of scientific evidence [32]. On the contrary, MMH-NPM is based on the European Pledge, which is a food industry initiative, including the soft drink industries (Coca-Cola, PepsiCo) and fast food companies (Burger King, Mcdonald’s) [42].

Among other interesting findings, which were not part of the main objectives, was that the most promoted brand was Coca-Cola, a brand that represented 12% to 13% of PP advertising transmitted in 2014 in the USA [37] and a company that is committed to not advertising F&B directed to children under 12 years of age [42]. Likewise, according to the US Federal Trade Commission, carbonated beverage companies invest millions in PP, spending 64% on PP aimed at adolescents [43].

The limitations of the study are (1) that only one researcher reviewed the programs, (2) the small sample size, and (3) that we could not record the programs. However, it should be considered that 39 programs could be viewed on several occasions on streaming platforms and official web pages (of each channel), a situation that should be analyzed, since PP advertisements in programs and films can be viewed repeatedly. To add, the use of PP in streaming platforms such as Blim, YouTube, and Netflix should be considered for future investigations and regulations [8,33].

## 5. Conclusions

PP was used to advertise nutrient-poor, energy-dense F&B in Mexican TV programs directed to children. The Mexican regulation does not adequately protect children against the effects of advertising because it is too permissive and does not cover all communication media and marketing strategies, such as PP. Therefore, the authors recommend: (1) including mass media, streaming platforms, radio, web pages, social networks, and advertising strategies like PP in the current regulation, as well as future forms of promotion; (2) defining nutritional criteria based on PAHO-NPM; (3) regulating the hours with the highest children audience; (4) creating policies based on evidence, in accordance with recommendations from international organizations, such as PAHO and WHO, while also considering the opinion of experts in the nutrition and public health area who are free of conflict of interest.

## Figures and Tables

**Table 1 ijerph-17-03086-t001:** Distribution of the analyzed hours.

Season	Weekend Days *	Weekdays *	Channel	Hours
Holidays	Saturday 17th of December 2016	Tuesday 27th of December 2016	2	6
			5	6
			7	6
			13	6
School	Saturday 7th of January 2017	Tuesday 17th of January 2017	2	6
			5	6
			7	6
			13	6
			Total	48

* Three analyzed hours, from 19 h to 22 h.

**Table 2 ijerph-17-03086-t002:** Characteristics of the analyzed Mexican public television programs, broadcast from 19 to 22 h.

Variables		Programs (n = 39)
		n (%)
Channel	XHTV Channel 2	10 (25.6)
	XHGC Channel 5	9 (23.1)
	XHIMT Channel 7	9 (23.1)
	XHDF Channel 13	11 (28.2)
Season	Holidays	19 (48.7)
	School	20 (51.3)
Genre	Soap operas	10 (25.6)
	Dramatic series	1 (2.6)
	Series	2 (5.1)
	Movies	2 (5.1)
	Reality shows	3 (7.7)
	Contests	1 (2.6)
	Comedy shows	4 (10.3)
	Talk shows	1 (2.6)
	Sports	2 (5.1)
	News	4 (10.3)
	Food TV shows	1 (2.6)
	cartoons	1 (2.6)
Product Placement	PP presence	10 (25.6)
	PP absence	29 (74.4)
Target Audience	Children	9 (23.1)
	General	30 (76.9)

**Table 3 ijerph-17-03086-t003:** Percentage of food and beverages advertised by product placement on Mexican public television, broadcast from 19 to 22 h.

Variables		Food and Beverages (n = 119) n (%)
Channel	XHTV Channel 2	-
	XHGC Channel 5	8 (6.7)
	XHIMT Channel 7	42 (35.3)
	XHDF Channel 13	69 (58)
Season	Holidays	73 (61.3)
	School	46 (38.7)
Genre	Soap operas	3 (2.5)
	Dramatic series	-
	Series	-
	Movies	46 (38.7)
	Reality shows	43 (36.1)
	Contests	-
	Comedy shows	-
	Talk shows	-
	Sports	25 (21)
	News	-
	Food TV shows	2 (1.7)
	Cartoons	-
Food Category	Savory and sweet snacks	11 (9.24)
	Sugar-sweetened beverages	49 (41.2)
	Dairy products	11 (9.24)
	Cereals	18 (15.1)
	Sauces, dips, and dressings	8 (6.7)
	Meat, poultry, fish, and similar	7 (5.9)
	Ready-made food	4 (3.4)
	Fruit, vegetables, and legumes	6 (5)
Type of Product Placement	Active	29 (24.4)
	Passive	90 (75.6)
Target Audience	Children	89 (74.8)
	General	30 (25.2)

(-) without product placement advertising.

**Table 4 ijerph-17-03086-t004:** Energy and nutrient content per 100 g/mL of food and beverage advertised by product placement on Mexican public television, broadcast from 19 to 22 h.

		Nutritional Content per 100 g/mL					
Categories	Variables	Energy (Kcal)	Sugar (g)	Fiber (g)	Total Fat (g)	Saturated Fat (g)	Sodium (mg)
	Total	154.9	2	0	1	0	21.88
Season ^†^	Holidays	214.5	1.2	0	2.1	0.14	29.13
	School	43.76	3.3	0	0	0	11.01
	*p*	< 0.001	0.480	0.103	< 0.001	< 0.001	0.506
Type of Product Placement ^†^	Active	62	3.3	0	0.05	0	10.21
	Passive	160	2	0	1.7	0.1	21.88
	*p*	0.919	0.983	0.229	0.247	0.278	0.468
Genre ^‡^	Movies	62 ^b^	5.5	0	0.26 ^b,c,d^	0	16.8
	Reality shows	259.4 ^a,c^	1	0	2.7 ^a,c,d^	0.33 ^c,d^	66.7
	Sports	43 ^b^	6.24	0	0 ^a,b^	0 ^b^	21.88
	Others	46	0	0	0 ^a,b^	0 ^b^	4
	*p*	<0.001	0.0547	0.098	< 0.001	<0.001	0.160
TargetAudience ^†^	Children	214.5	1.2	0	2.1	0.14	21.88
	Other	43	2.93	0	0	0	21.88
	*p*	<0.001	0.763	0.110	<0.001	<0.001	0.936

n: 119 advertisements; ^a^ Movies, ^b^ Reality shows, ^c^ Sports, ^d^ Others (Food TV shows and Soap operas); ^†^ Mann–Whitney U test; **^‡^** Kruskal–Wallis test with Bonferroni correction.

**Table 5 ijerph-17-03086-t005:** The percentages of foods and beverages classified as not healthy according to three nutritional profile models.

Food Category		MMH-NPM ^a^		WHO-Europe ^b^		PAHO-NPM ^c^	
	Total	n	%	n	%	n	%
Savory and sweet snacks	11	9	81.8	11	100.0	11	100.0
Sugar-sweetened beverages	49	40	81.6	49	100.0	49	100.0
Dairy products	11	10	90.9	4	36.4	11	100.0
Cereals	18	2	11.1	1	5.6	3	16.7
Sauces, dips, and dressings	8	4	50.0	6	75.0	8	100.0
Meat, poultry, fish, and similar	7	0	0.0	0	0.0	0	0.0
Ready-made food	4	4	100.0	4	100.0	4	100.0
Processed fruit, vegetables, and legumes	6	5	83.3	4	66.7	6	100.0
Total	119	74	62.2 ^c^	79	66.4	92	77.3 ^a^

^a^ MMH-NPM, ^b^ WHO-Europe, ^c^ PAHO-NPM, letters in superscript indicate difference with the group (*p* < 0.05).

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
