# Peer review of "Nutritional Quality of Hidden Food and Beverage Advertising Directed to Children: Extent and Nature of Product Placement in Mexican Television Programs"

_ijerph, 2020, doi:10.3390/ijerph17093086_

Round 1

Reviewer 1 Report

Review report

Manuscript ID: ijerph-767315

A brief summary

The aim of the study was 'to characterize the nutritional quality of foods and beverages advertised by product placement in Mexican television programmes with the highest children's ratings' using three various nutrition criteria and to analyse the nutritional value of the advertised foods and beverages according to season, type of product placement, programme genre and target audience. This research is very important to nutrition science in the times when mass media play central role in the daily life and are capable to strongly influence human choices, including food choices.

Broad comments

The presented research is valuable and its outcomes should result in working out better regulations for protecting children from the adverse effects of advertising unhealthy foods. However, there are some substantial improvements which the Authors should make.

The main problem is the language - many parts of the manuscript are difficult to read and the reader has to make much effort to understand what the Authors aim to express (sometimes it is impossible anyway). Many formulations are unnatural to English language, some are probably a direct translation from the Authors' native language, many are incorrect, sometimes singular is used instead of plural and vice versa. Therefore, the whole text should be rewritten by an English native speaker.

The section Results should be rewritten. Jargon and incorrect formulations are used.

Also the section Discussion should be rewritten. The Authors divided the text into very short paragraphs and failed to achieve the continuity of the text. Those paragraphs which discuss the same problem should be integrated into one coherent paragraph showing clearly the Authors' idea.

The Authors should also think over once again whether their comparisons of the results to other researchers' results are appropriate. For example, see lines 292-294: the obtained results are described as similar to the results obtained in Ireland and the UK where 47.5% of foods and 25% of SSBs were considered unhealthy - in my opinion the results obtained by the Authors' are much worse.

The section Conclusions should be rewritten. In this section, the Authors should briefly highlight the main conclusions from their study. Therefore, no references should be used as well as the statements which cannot be drawn from the results obtained by the Authors.

Specific comments

Instead of 'directed at' (in the title and throughout the manuscript), 'directed to' should be used (please check up a monolingual dictionary).

References should always be placed before punctuation marks, e.g. 'those rates increased since 2000 [2].'

Pay attention to capital letters, e.g. Mexican Ministry of Health (line 19), the Ministry of Health (line 125), Mexican population (line 357).

'Et al.' instead of 'and cols' should be used (line 96) (or 'and colleagues', if so).

After the names of researchers, a reference should be placed.

The text below the title '2. Material and Methods' should have a subtitle and number 2.1, and so the subtitle 'TV programme information' should have number 2.2 and so on. The same refers to numbering section 3 and its subsections.

What is A, AA, B etc. classification (lines 162-163)? The Authors should explain it, especially taking into account that the reference is in Spanish.

'Content of energy...' should be used instead of 'Data of energy...' (line 177).

38.7% is not the majority (line 231) - the majority is more than a half (especially much more than a half). 38.7% is the highest percentage.

Line 249 - 'median energy and nutrient content... ' while in table 4 the comment by the asterisks informs about mean values.

Lines 250-251: 'Significant differences were observed in season, genre and target audience' - it is an example of using a jargon in this manuscript. Indeed, statistically significant differences were observed in energy and nutrient content according to season, genre and target audience. Pay attention to forming other statements referring to the results.

Lines 304-305: example of an inappropriate formulation; the Authors meant that the products advertised during holiday were characterised by higher energy and fat content.

Line 323: this sentence is incomprehensible.

Table 4 - Maybe it would be easier to read the table if there was 'Energy and nutrient content per 100 g' instead of 'Nutritional content' and the asterisks.

Table 5 - please, rewrite the title of the table, e.g. 'The percentages of foods and beverages classified as unhealthy according to three...'

Reviewer 2 Report

The authors conducted a study that focuses on product placement in Mexican shows and the nutritional quality. Even though this is an interesting study, the following modifications are suggested to strengthen the manuscript:

Title: clarify if it was children, 5-11 years of age, or youth, 5-19 years of age?  That was the focus.

Abstract:

Instead of “unhealthy” recommend poor nutrient-dense foods. In the objective did not mention nature or extent as was mentioned in the title, just nutritional quality. In the results, sugar-sweetened beverages are not a food category, please revise to include the food category that was represented the most in that 60%. Interesting that even though the sugar sweetened beverages were represented the most, added sugars was not found to be statistically significant. Please clarify it is was commercials that was viewed during this 48-hour period or just observations of food/beverage consumption in television shows as the conclusion statement said stricter regulations for advertisement. Therefore, is the goal to eliminate all food/beverage consumption in these shows or just have more nutrient-dense foods that the actors and others consume?

Introduction:

Please use more scientific terminology, so instead of “nowadays” possibly “currently”. Also, remove unhealth and replace with poor nutrient-dense foods/beverages.

Limited information was provided about identifying nutritional quality of these products based on the information that was discussed, suggest including that information earlier in the introduction.

Line 35: Instead of Early, maybe this was supposed to be every?

Another reason youth are consuming poor nutrient-dense foods is because of what parents purchase. Even though we could say it is advertisement, the home-environment also has something to do with this as if parents prefer sugar-sweetened beverages over water, the child will grow up not recognizing that water is preferred over these other beverages. It is something else to include as these advertisements may have something to do with it, but it a socio-ecological issue.

Line 61: change salt to sodium and remove ‘drastically out of line’ to a more technical term.

Lines 71-79: should be integrated earlier to discuss advertisements, then discussing how Mexican children are exposed to poorer nutrient-dense foods/beverages.

Methods:

Even though those 4 dates were randomly chosen, there was limited explanation of why Tuesday of a weekday? Please include that information.

Did only 1 researcher identify the products, please clarify. Also, indicate the training that took place. Indicate if another researcher assisted in the validation of the product placement. If a product was shown three times throughout the show, was this captured? Also, the length of time that product was on display? Please clarify this information. Was each show recorded due to the number of channels that were reviewed and then the analysis was completed later, please clarify.

Was data not analyzed on sodium? Please clarify. Please expand on the instruments used to assess nutritional quality for readers not familiar with these.

Discussion:

Did not really discuss why products were placed, but instead just mentioned how it was consistent with other studies. Need to associate this more then back to why overweight/obesity exists and how these poor-nutrient dense foods contribute to this issue.

Reviewer 3 Report

Dear authors,

Your manuscript describes an interesting area. My only question/suggestion is related to the use of nutrition content of the products (lines 173-179) and the results presented in table 4. I do not understand what these analyses and results would add to the research question. It only counts total amounts, but it is not clear from which products, and does not say anything about the amount of products or nutrients regularly consumed (it is now calculated per 100g of product). So to me, this does not give relevant information. I would advice to delete these parts of the manuscript.

Round 2

Reviewer 1 Report

Review report

Manuscript ID: ijerph-767315

The manuscript has been largely improved. I still have some remarks which should be included to make the manuscript meet the standards of the Journal.

Line 59-60: 'F&B most frequently promoted to children are high in saturated fat, sugar, or sodium, according to dietary recommendations' - please, delete the phrase 'according to dietary recommendations' because it sounds as if this high content of saturated fat etc. was in line with the recommendations.

Line 68 and throughout the manuscript: you should use either 'targeted at' or 'targeted towards' instead of 'targeted to' (please, check up the Oxford Collocations Dictionary).

Line 77: '... the audience [19]. For example... ' - please, change the period into a comma: '... the audience [19], for example... ' because the phrase after 'for example' is indeed the part of the previous sentence.

Line 96: 'Et. al.' - please, change to 'et al.', that is without a point after 'et' (it is from Latin 'et alii' meaning 'and others') and starting with lowercase (I wrote in my report starting with a capital letter because it was at the beginning of the sentence).

Line 99-100: it is better to place the reference [24] after the surname 'Sutherland'.

Line 102: 'Uribe R and Fuentes-Garcia A' - you should omit the name initials, that is 'R' and 'A'.

Line 104: 'In the case of age' is in my opinion a jargon; you should either change it or simply start the sentence 'Older children (12-15-year-old)… '

Line 105: start 'brand' using lowercase.

Line 106: always use Simple Past when describing findings of a research, that is use 'increased' instead of 'increases'.

Line 111: 'Mexico regulated the advertising of F&B advertisements directed to children' - it is not a good formulation: 'advertising of ... advertisements', therefore, I suggest: 'Mexican government imposed regulations of F&B advertisements directed to children', however, please consult it with an English native speaker.

Line 112: 'in TV and cinemas' - please, change to 'broadcast on TV and shown in cinemas'.

Line 119-120: 'In addition to the these limitations' - please, delete 'the'.

In my opinion, the sentence in line 137-138 ('We selected a convenience sample...') became unnecessary after you had added the sentence in lines 135-136. You should only leave the information that these days are described in table 1.

Title of the table 1: please add 'the' before 'analyzed'.

Line 159: please, delete the point after 'T', use the American English version of the word 'program' (because you have changed the spelling in the whole manuscript from British English to American English) and start the word 'program' with lowercase.

Line 165: if you decided to use plural in the case of soap operas, series etc., so you should consequently use plural also in the case of talk shows, food TV shows and cartoons. Please, check the whole text, especially tables 2-4.

Line 181: start 'nutrient' with lowercase.

Lines 188-189: 'the USDA Food Composition Databases was used' - please, correct: 'the USDA Food Composition Databases were used'.

Line 194: you lost 't' at the end of 'nutrient'.

Line 197: start 'not' with lowercase.

Title of table 2 - please add 'the' before 'analyzed'.

Line 236: the past form of 'broadcast' is also 'broadcast' as it is an irregular verb.

Line 270-271: In my opinion, the sentence 'By type of product placement, there were no statistically significant differences' is unnecessary but if you want to write it, it should be at the end of the paragraph.

Line 272: 'during the reality show genre are higher in energy' - please, change to 'during the reality shows were higher in energy'.

Line 276: there should be Simple Past Tense: were not targeted at them.

In table 4, after the column caption 'Nutritional content', please add 'per 100 g/ml'

Below table 4: please, change the footnote by the symbol '‡' - it is better to write: 'Kruskal-Wallis test with Bonferroni correction'.

Line 288: if it is unavoidable, start the sentence with the value (62.2%); do not add 'about' because the value is precise, with decimal value after the point.

Line 290: 'represented' is not a suitable word; I suggest: 'comprised'.

Line 294: please, delete 'the' before '100%'.

Line 305: always use Simple Past when describing findings of a research, that is use 'were' instead of 'are'.

Line 312: please, omit 'however'. This word is used to introduce a statement that contrasts with something that has just been said. You can use 'The results of the studies... ' to start the new sentence.

Line 313: instead of 'found similar results' I suggest 'are slightly less alarming' because the percentages of unhealthy foods and SSB were lower than in the presented results.

Line 314: please, omit 'that were'. It is not necessary and you have less 'were' in this sentence making it easier to read.

Line 316: the reference [8] before the period.

Line 326: it would be good to start a new paragraph with the sentence 'In relation…', however, I suggest rewriting this sentence: 'As far as type of program is concerned, movies were the principal genre in which PP advertisement of energy-dense nutrient-poor F&B was used'.

Line 328: 'the study in the USA analyzed' is not appropriate; I suggest 'Also the analysis of the prevalence of F&B... most viewed movies in the USA for each year… to 2005 showed that... '

Line 330: please, again use Simple Past, divide the sentence and move the sentence from line 332 that is: 'Another relevant genre were reality shows most probably because reality shows are the second genre with the highest children audience in Mexico [10]. There is evidence that... '

Line 335: I suggest: 'holiday season [14], contrary to our findings on PP advertising.

Line 340: you should emphasize: ' The presented results... ' or 'Our results... '

Line350: there should be a reference right after 'other study in Mexico'.

Line 350: 'the study compared' is a jargon; a person may compare, so I suggest 'Rincón-Gallardo Patiño et al. [15] compared... '

Line 354: please, use 'had' instead of 'has'.

Line 356-357: 'on the other side' does not mean in English what you mean; I suggest to write: 'On the contrary, Mexican criteria... '

Line 359: correct the spelling of McDonald's.

Line 362: instead of 'brands were' there should be 'brand was' because only one brand is mentioned.

Line 363: you should add 'is' before 'committed' that is 'a company that is committed to'.

Line 375: your conclusion refers to your findings which you should emphasize, that is: 'PP was used to advertise nutrient-poor energy-dense F&B in Mexican TV programs directed to children'.

Line 377: please, delete the reference - in the conclusions you should refer only to your findings.

Line 381: in my opinion, you should write 'children audience' instead of 'children ratings'.

All the best for your current and future research!

Reviewer 2 Report

The authors have significantly revised this manuscript, thus the reviewer has no additional comments.

Author Response

We appreciate a lot the time and detail you gave to the revision of the document.